# A Review of the Multi-Systemic Complications of a Ketogenic Diet in Children and Infants with Epilepsy

**DOI:** 10.3390/children9091372

**Published:** 2022-09-10

**Authors:** Kyra Newmaster, Zahra Zhu, Elizabeth Bolt, Ryan J. Chang, Christopher Day, Asmaa Mhanna, Sita Paudel, Osman Farooq, Arun Swaminathan, Prakrati Acharya, Wisit Cheungpasitporn, Siddharth Gupta, Debopam Samanta, Naeem Mahfooz, Gayatra Mainali, Paul R. Carney, Sunil Naik

**Affiliations:** 1College of Medicine, Penn State University, Hershey, PA 17033, USA; 2College of Medicine, University of Toledo, Toledo, OH 43614, USA; 3Faculty of Medicine, Al-Quds University, Jerusalem P.O. Box 51000, Palestine; 4Department of Pediatrics and Neurology, Penn State Health Milton Hershey Medical Center, Hershey, PA 17033, USA; 5Department of Neurology, University at Buffalo, Buffalo, NY 14202, USA; 6Department of Neurological Sciences, University of Nebraska Medical Center, Omaha, NE 68198, USA; 7Division of Nephrology, Texas Tech Health Sciences Center El Paso, El Paso, TX 79905, USA; 8Department of Medicine, Mayo Clinic, Division of Nephrology and Hypertension, Rochester, MN 55905, USA; 9Kennedy Krieger Institute, Department of Neurology, Johns Hopkins University, Baltimore, MD 21287, USA; 10Child Neurology Section, Department of Pediatrics, University of Arkansas for Medical Sciences, Little Rock, AR 72202, USA; 11Pediatric Neurology Division, University of Missouri Health Care-Columbia, Columbia, MO 65201, USA

**Keywords:** ketogenic diet, modified atkins diet, epilepsy, children, pediatrics, seizures, nutrition

## Abstract

Ketogenic diets (KDs) are highly effective in the treatment of epilepsy. However, numerous complications have been reported. During the initiation phase of the diet, common side effects include vomiting, hypoglycemia, metabolic acidosis and refusal of the diet. While on the diet, the side effects involve the following systems: gastrointestinal, hepatic, cardiovascular, renal, dermatological, hematologic and bone. Many of the common side effects can be tackled easily with careful monitoring including blood counts, liver enzymes, renal function tests, urinalysis, vitamin levels, mineral levels, lipid profiles, and serum carnitine levels. Some rare and serious side effects reported in the literature include pancreatitis, protein-losing enteropathy, prolonged QT interval, cardiomyopathy and changes in the basal ganglia. These serious complications may need more advanced work-up and immediate cessation of the diet. With appropriate monitoring and close follow-up to minimize adverse effects, KDs can be effective for patients with intractable epilepsy.

## 1. Introduction

Epilepsy in children can vary in presentation and severity, requiring a multi-faceted approach to treatment. Up to 65% of patients attain spontaneous remission or are well-controlled on anti-seizure medications, but a significant number of patients continue to have recurrent seizures despite attempting numerous medical therapies alone or in multi-drug regimens [1,2]. These refractory patients can be managed with other choices such as curative or palliative epilepsy surgery. Examples include neuromodulators such as vagal nerve stimulators (VNS), responsive neurostimulation (RNS), and deep brain stimulation (DBS) [3]. Patients looking for non-surgical options turn to complementary medicinal options such as herbal remedies, supplements, or cannabis-related products [4]. Finally, a frequent option for the control of refractory epilepsy is the ketogenic diet (KD) [5].

KDs are considered viable for patients who are unresponsive to two or more anti-seizure medications [5]. Some parents prefer to start a KD earlier because they perceive it as a “natural” option and expect it to come with fewer side effects. KDs are generally well-tolerated, making this class of therapy efficacious and versatile in the control of both idiopathic and congenital epileptic syndromes such as GLUT1 Deficiency Syndrome [5,6,7].

A KD is a dietary therapy that induces a ketotic state through carbohydrate restriction. Ketosis is a metabolic state where cells rely on ketone bodies and fatty acids once glycogen stores have been depleted. Within the brain, ketone bodies modulate neurotransmission and provide additional anti-inflammatory effects [8,9]. Specifically, ketone bodies increase GABA while simultaneously decreasing glutamate reuptake [10]. Finally, low carbohydrate states alter ATP production and inhibit ion channel activity which reduces the metabolic demand of neurons providing neuro-protection during seizures [10].

There are four different ketogenic diets that have been used for medical purposes [11,12]. These include the classical ketogenic diet (cKD), the traditional medium chain triglyceride ketogenic diet (tMCTKD), the modified medium chain triglyceride ketogenic diet (mMCTKD), and the modified Atkins diet (MAD). Additionally, the low glycemic index diet (LGIT) is not a true KD, but it is a carbohydrate restricted diet that is efficacious in epilepsy management [11,13]. The cKD is predominantly composed of long chain triglycerides (LCTs), and the fat to carbohydrate ratio ranges from 3:1 to 4:1 [14,15]. Carbohydrate restriction is also sometimes combined with additional calorie or protein restrictions [16,17]. Traditionally, the cKD was started with a medically supervised fast in the hospital to help the patient achieve a baseline level of ketosis. Some patients find it helpful in transitioning to a KD from a behavioral standpoint, but this is no longer deemed necessary. The tMCTKD and mMCTKD were designed as more flexible alternatives to the cKD. Commercial medium chain triglycerides (MCTs) produce more ketone bodies per kilocalorie than LCTs, allowing for a higher carbohydrate intake [18,19]. However, all three diets are still fairly restrictive.

The MAD and LGIT diets were designed to allow more dietary freedom. While the previous diets have a 4:1 or 3:1 fat to carbohydrate ratio, the MAD has a ratio of 2:1 to 1:1 [11,20,21]. On the other hand, the LGIT is not a true KD as it simply aims to prevent large increases in postprandial blood glucose by focusing on low glycemic index foods and limiting total daily carbohydrates [16,22]. In studies by El-Rashidy and Lakshminarayanan, both types of diets were shown to have similar efficacy in seizure control when compared to the cKD, tMCTKD, and mMCTKD [23,24].

KDs are also widely used to treat non-epileptic conditions such as obesity, headaches, migraines, autoimmune diseases, and cancer due to their anti-inflammatory and metabolic effects [10,25,26,27]. Despite the versatility of KDs, they are not completely benign. Therefore, it is important to carefully document complications that result from this type of therapy and what can be done to mitigate these problems [28,29,30,31,32,33]. Here, we provide a comprehensive review of different KD complications and clinical solutions in pediatric patients which is summarized in Table 1.

## 2. Complications of a Ketogenic Diet by System

### 2.1. Gastrointestinal Complications

According to a 2017 meta-analysis, gastrointestinal (GI) complications including constipation, diarrhea, vomiting, and gastroesophageal reflux are the most commonly reported side-effects, with up to 40% of reported side-effects being GI related. However, these problems occurred primarily at the initiation of the diet [34]. Within the GI category, constipation, nausea, and vomiting are the most reported, and are easily managed with polyethylene glycol and ondansetron [29].

A more serious GI complication that may require modification or cessation of the KD is hypoproteinemia, which can be caused by enteropathy secondary to intestinal inflammation or lymphatic leakage of protein. Enteropathy in the context of a KD often occurs when another underlying pathology exists [35]. For example, a case report by Wang et al. describes a three-month-old female with a genetic epileptic syndrome (STXBP1), managed on a 4:1 ketogenic diet, who presented with an intestinal lymphangiectasia and protein-wasting enteropathy [36]. This young female’s enteropathy was alleviated by decreasing her fat to carbohydrate level to a 1:1 ratio, and despite this dietary change, she remained seizure free for at least 20 months. Another possible trigger for KD-induced enteropathy is previously undiscovered intolerances and allergies, but there is only one report of a protein-losing enteropathy during the ketogenic diet, due to a soy allergy [37]. Furthermore, Kang et al. and Suo et al. also found unexplained hypoproteinemia in 12 of 129 patients (9.3%) and 39 of 317 patients (12.3%) respectively, indicating that subclinical enteropathy or other factors may be present in a larger population than what is documented [30,38]. These reports highlight the importance of carefully considering all of the patient’s medical history in order to craft an individually appropriate KD. Furthermore, it may be helpful to consult a GI specialist and if necessary, utilize scintigraphy and colonoscopy, which have been previously used in KD treated patients with hypoproteinemia to diagnose suspected enteropathy [39].

Other rare extraluminal GI effects of the KD diet include cholelithiasis, pancreatitis, and hepatitis [40,41,42]. Cholelithiasis is most likely triggered by metabolic changes and increased gallbladder activity. Only two cases of this complication have been reported and only one of them needed surgical management [40,41]. Pancreatitis appears to be a little more common and is thought to be a result of increased serum lipids [42]. Most reports of this side effect occurred in obese or overweight patients embarking on a KD for weight loss purposes, who likely had underlying dyslipidemia. Pediatric patients do not commonly have this problem, but three pediatric cases have been reported [42]. One patient was discussed in a case report and the other two were reported in a retrospective study of 71 patients [42]. Of these, one patient died due to necrotizing pancreatitis [43]; the other two simply stopped the diet and the pancreatitis remitted [44]. The patient who passed away had an EIF2S3 mutation that triggered seizures and pancreatic dysfunction [43]. Hepatitis is a rare complication of the KD, but concomitant use of valproic acid (VPA) has been thought to increase the risk of hepatitis. However, Kang et al. found that there was not a significant increase in the number of hepatitis cases in patients taking VPA and following a KD [45]. Additionally, low carnitine levels in some KD patients may also contribute to liver dysfunction [45]. Hence, screening for familial dyslipidemia and pancreatic dysfunction along with routine monitoring of blood lipids, liver enzymes, and carnitine levels is recommended [46].

### 2.2. Cardiac Complications

KD-induced cardiac complications are rare, with the most serious concerns being a prolonged QT interval and cardiomyopathy [30,46,47,48]. In the 2000s, QT prolongation was reported in at least three children whose epilepsy was being managed with a KD [49,50]. Two of these children suddenly passed away after developing torsade de pointes [49]. Furthermore, the length of the QT interval was inversely related to the bicarbonate concentration and directly correlated to ketone concentrations, suggesting that the cardiac abnormalities were a result of metabolic disturbances [50]. However, more recent prospective studies that included at least 80 patients showed that these children did not have any change in QTc length from their baseline after at least one year on a KD, raising questions about how common this condition is [48,51].

KD-related cardiomyopathy occurs in the context of selenium deficiency, which is typically a rare cause of cardiomyopathy [46,51,52,53]. A 2003 report of 39 epilepsy patients on a KD was triggered by a single patient with no detectable selenium in their blood [47]. This study showed that 9 of the 39 patients had significantly reduced serum selenium. However, only the index case had evidence of cardiomyopathy by echocardiogram [47]. Others have also reported selenium deficiency both with and without cardiomyopathy [54,55]. In 2015, a prospective study with 61 patients on a KD documented cardiac outcomes for one year [51]. Measures for this study included serum carnitine, serum selenium, electrocardiography, and echocardiographic examinations throughout the study. None of these measures were significantly different between the start and the end of the trial, except for the mitral A wave (0.66 m/s vs. 0.61 m/s, *p* = 0.05) [51]. Others have challenged the presence of selenium deficiency in KD patients, but often these studies last for approximately one year, which may not be enough time to develop this complication [30,48]. Ultimately, a protein restricted or deficient diet may result in selenium deficiency as animal proteins tend to have the highest concentration of selenium. Therefore, selenium concentration monitoring and selenium supplementation should be considered in patients who are not eating significant amounts of animal products.

### 2.3. Vascular Complications

Because KDs are high in fat, there is concern that increased dietary fats could result in dyslipidemia and atherosclerosis. Kapetanakis and colleagues evaluated 26 epileptic children treated with a KD and found decreased carotid distensibility and worsened lipid profiles at 3- and 12-months post KD initiation, but the changes stabilized at 24 months [55,56]. Another study by Ozedmir and colleagues found that an olive oil-based KD does not appear to change aortic and carotid artery elasticity in pediatric epilepsy patients. However, this type of KD was still associated with an increased concentration of serum lipids, consistent with the work of Kapetanakis [55,56]. Given the propensity of the KD to increase serum lipid levels, patients who have existing metabolic dysregulations must be monitored carefully.

### 2.4. Renal Complications

Renal calculi have been reported in 3–7% of children on a ketogenic diet, consisting of both uric acid and calcium [13,57,58,59,60]. In addition, three studies of over 200 children on a KD, showed that potassium citrate led to a reduced incidence of renal calculi [58,61,62]. Dressler and colleagues also successfully treated kidney stones in infants on a KD by alkalizing their urine with basic salts such as Polycitra-K^®^ and Uralyt-U^®^ [63].

There is also a theoretical increased risk of renal calculi in patients on a KD and anti-epileptic drugs (AEDs) with carbonic anhydrase inhibitor activity such as zonisamide and topiramate [64]. However, one study showed that regardless of the patient’s medication status, urolithiasis is associated with the presence of metabolic derangements including metabolic acidosis, concentrated urine, acidic urine, hypercalciuria and hypocitraturia [64]. Finally, in rare instances, hematuria without the presence of a stone has also been reported [44]. Therefore, a risk–benefit analysis of initiating a KD in patients with preexisting kidney abnormalities, such as Alport syndrome, polycystic kidney disease or nephropathic cystinosis, must be done. A renal ultrasound may also be recommended before the initiation of a KD and at six months to one year after starting a KD, to diagnose asymptomatic renal stones.

### 2.5. Hematological Complications

Hematological complications of the KD include neutropenia, increased hemoglobin, and decreased platelet function [64,65,66]. In a retrospective study by Munro and colleagues, 27 of 89 children developed neutropenia, which was associated with increased urinary ketones and longer duration of the KD. Despite the presence of neutropenia, there were no clinically significant infections [67]. Interestingly, other studies that investigated complete blood counts showed a normal neutrophil count [64,65]. Some reports also found significantly increased hemoglobin, hematocrit, MCV, and serum vitamin B12, with no apparent clinical manifestations [64].

Finally, platelet dysfunction and bruising have also been reported in some patients on the KD [66]. Further study of six patients revealed increased bleeding time and diminished platelet reactivity to aggregating factors, suggesting altered membrane dynamics secondary to lipid changes [66]. However, a more recent study of 162 children on a KD for epilepsy found no differences in platelet counts or function after one year on the KD [68]. In this study, it was also shown that desmopressin was able to stimulate clotting in vitro, suggesting a possible treatment for patients with concerning bruising [66].

### 2.6. Neurological Complications

The KD is generally thought to be a neuroprotective treatment, and only one major negative neurological outcome has been reported. In 2003, Erickson and colleagues reported on a child with cryptogenic epileptic encephalopathy who developed chorea and ataxia corresponding with putaminal lesions on MRI three weeks after starting a classical KD [69]. A subsequent MR spectroscopy study showed a lactate peak in the basal ganglia, suggesting a failure of mitochondrial energy metabolism. Both the MRI and MRS abnormalities resolved after discontinuing the KD, while her new onset movement disorder did not. Since this case report, no other neurological sequelae have been reported in the use of KD for epilepsy. The authors of this report also suggested that a secondary mitochondrial condition such as familial striatal necrosis could be the culprit [69,70].

### 2.7. Dermatological Complications

Prurigo pigmentosa is a papular, pruritic rash that appears rapidly after the commencement of a KD, and it responds well to increased carbohydrate intake or tetracyclines [71,72]. Originally, this rash was thought to be specific to East Asian populations [72], but later it was reported in two Middle Eastern patients on strict diets with prolonged fasting [73]. Furthermore, a recent literature review identified 19 case reports of prurigo pigmentosa in patients following ketogenic diets. The highest proportion of reports were among Asian (32%), Middle Eastern (32%), and Caucasian (11%) patients; this report also found one African American and one Hispanic patient with this complication [74]. This distribution suggests that there is no genetic association for this symptom, and no literature to date reports such an association.

### 2.8. Bone Density and KD

Young patients on a KD often have osteopenia due to reduced vitamin D and calcium levels, which predisposes patients to pathological fractures [75,76,77]. Non-ambulatory patients and patients with a BMI greater than 25 are more susceptible to poor bone health [78]. Therefore, it is important to have a high level of suspicion for osteopenia in these patients. Serial DEXA scans every 1–2 years are helpful to screen for early signs of osteopenia, and treatment with bisphosphonates in patients with low bone density can be used to prevent fractures. Vitamin D and calcium supplementation also appears to be helpful in preventing osteopenia [78].

### 2.9. Antiseizure Medications and KD Interaction

Generally, it has been shown that KDs do not affect the level of AEDs in the blood [79,80,81,82]. However, a recent prospective study of 63 epileptic patients who were on the MAD and taking AEDs showed reduced clobazam 12 weeks after diet initiation [83]. This may be attributed to specific food content in the MAD which can induce AED metabolizing enzymes [84]. On the other hand, due to their lipophilic nature, AEDs such as carbamazepine, phenytoin, and valproic acid (VPA) could also be absorbed more readily with a high fat diet [85,86]. Phenobarbital has also been shown to increase by up to 100% in patients on a ketogenic diet, and it more readily crosses the blood brain barrier in patients on a KD, which has been shown to trigger altered mental status at normally tolerated doses [87]. Therefore, it is important to avoid or reduce phenobarbital doses and monitor AED levels in patients on a KD. It is also important to note that AEDs can also impact the efficacy of a KD. For example, VPA and lamotrigine have been shown to inhibit ketosis and precipitate carnitine deficiency [88,89,90]. Therefore, it may be worthwhile to supplement with carnitine in patients taking these AEDs.

### 2.10. Surgical Considerations

Because many medical interventions during surgical procedures can introduce carbohydrates into the bloodstream, special considerations have been suggested for surgical procedures in KD dependent patients. However, patients on a KD do not appear to have increased risk for anesthesia-related complications according to several retrospective studies, which collectively reported only one incident [91,92]. However, metabolic acidosis is reported more often and has been attributed to the use of IV fluids, namely lactate ringers, which contain electrolytes and worsened the pre-existing metabolic acidosis in KD patients [93]. Therefore, it is important to use normal saline while monitoring pH and bicarbonate levels during surgical procedures and in patients requiring IV fluids [92]. Blood products must also be administered with caution due to their unknown glucose content [93]. Finally, efforts to avoid carbohydrate-containing medications such as propofol are important for preventing post-surgical ictal events in patients on a KD for epilepsy management [93,94].

### 2.11. COVID/Respiratory

COVID has an unclear interaction with the KD. However, there is some suggestion that a KD may be beneficial for patients infected with SARS-CoV2. Specifically, a KD high in MCTs may inhibit COVID replication by reducing envelope formation [95,96]. Furthermore, there are no indications that the KD is detrimental in SARS-CoV2 infections, and a low carbohydrate diet produces less carbon dioxide, which may be protective against respiratory failure [97]. Furthermore, KDs have been shown to be safe in conjunction with mechanical ventilation [98,99]. However, very low carb diets have been contraindicated in patients with respiratory failure, particularly if they are obese, due to concerns that excess body fat may worsen respiratory failure [97]. Therefore, management of critically ill patients must take into consideration all factors of health including underlying epilepsy.

### 2.12. Growth and Nutrition

While a KD can be a viable therapeutic option for pediatric epilepsy, the energy requirement of these patients must be considered to minimize disruptions in growth. The high fat content of KDs decreases appetite, which in theory would lead to inadequate caloric intake and reduced growth [100]. A study by Groleau and colleagues partly supports this idea. They demonstrated that children with intractable epilepsy on a long-term KD had decreased linear growth compared to non-KD counterparts, while their weight status and resting energy expenditure remained the same [101]. Similarly, a 2-year prospective study reported that 9% of the participants had growth that fell one standard deviation below the age-adjusted normal, even though they had normal nutritional status [102]. This suggests that caloric insufficiency is not the main driver of reduced growth in patients on a KD. A more likely culprit may be skeletal effects, which are described above, making bone density monitoring and vitamin D supplementation even more critical in pediatric patients.

KDs also disrupt micronutrient levels outside of those discussed above. For example, one case report of a nine-year-old female patient with refractory epilepsy who swallowed a primary molar and then subsequently developed scurvy highlights the possibility of developing vitamin C deficiency when following a KD [103].

Carnitine, an amino acid required to transport fatty acids into the mitochondria for energy production, can also be depleted early in the KD treatment, but carnitine levels typically stabilize with carnitine supplementation. Over time, supplementation is generally not needed unless the patient is symptomatic [104]. Nonetheless, these studies demonstrate the need for careful monitoring for vitamin and mineral deficiencies in pediatric epilepsy patients on a KD.

### 2.13. Metabolic and Endocrine Complications

Numerous studies have connected the KD’s high fat content to metabolic issues such as hyperlipidemia. As discussed above (see 2.3 Vascular Complications), numerous studies have shown significant increases in total cholesterol, LDL, VLDL, triglycerides, and total apoB lipoproteins six months after the initiation of the KD [105,106,107,108]. However, one study showed that there was not a significant change in plasma lipids at six weeks, which suggests that these findings take some time to develop [19]. Furthermore, the same study also found no change after one year, and another study by Yilmaz and colleagues demonstrated that the total cholesterol and triglyceride levels were increased for a period of time but returned to baseline after 24 months post-initiation, specifically in patients on a high olive oil intake [19,109]. These results also suggest that lipid levels may normalize over time. The role of fat sources in manipulating lipid levels remains equivocal.

Theoretically, a KD that is high in unsaturated fat may be beneficial for patients with pre-existing dyslipidemia. For example, Liu and colleagues have shown that some patients with pre-existing dyslipidemia can have a significant reduction in mean total cholesterol, low-density lipoprotein, and total cholesterol/high-density lipoprotein when treated with a KD utilizing a low amount of saturated fats [110]. Therefore, a low saturated fat KD could be considered for children with hyperlipidemia and refractory epilepsy.

Other metabolic complications of the KD include ketoacidosis and catabolic crises, which occur mostly in the presence of other underlying conditions. Patients with type 1 diabetes or those who take sodium-glucose cotransporter-2 inhibitors (SGLT-2i) have lower insulin activity. These patients have an increased propensity for developing ketoacidosis, which can be exacerbated by a low carbohydrate diet [111]. In addition, the use of ketogenic diets can cause a severe catabolic crisis in people with inborn errors of metabolism that impede the transport or oxidation of long-chain fatty acids [13].

Lipids also have critical interactions with endocrine functions and reproductive health. For example, one study showed that a KD may be beneficial for women with PCOS due to the significant reductions in glucose and insulin levels after twelve weeks of dietary intervention [112]. The lowered levels of insulin are thought to decrease IGF-1 and thus reduce aberrant androgen synthesis in both the ovary and the adrenal glands [112]. This has tremendous implications for developing children, and special care should be taken to look for gonadal dysfunction in young patients on a KD.

### 2.14. KD Complications in Infants

The first two years of life have the highest incidence of seizure disorders [113]. This is in part due to a number of genetic conditions that trigger seizures, many of which are resistant to medications. Thus, KDs are particularly useful treatment options for these patients. For example, a KD is indicated as the treatment of choice in infants with glucose transporter type 1 (GLUT1) and pyruvate dehydrogenase complex (PDC) deficiency syndromes [114]. Nordli and colleagues also showed that the KD was efficacious in treating infantile spasms and myoclonic seizures [115].

However, complications were reported in 6 out of 32 infants, including severe vomiting, renal stones, gastrointestinal bleeding, type I hyperlipidemia, ulcerative colitis, and coma. Most of the patients who developed the complications were on the diet for more than one year, and most side effects in infants were transient and reversible [115]. In a retrospective review of 29 infants treated with the KD, 75% of participants experienced either no side effects at all or very minor ones such as moderate vomiting or constipation [116]. Adverse biochemical events, such as hypervitaminosis E, hypercholesterolemia, or zinc level abnormalities were reported in 38% of patients within this study. However, none of the side effects were severe enough to stop the KD. A recent study of 171 infants on a KD reported adverse effects in 50% of the participants [117]. The most commonly reported side effects were dyslipidemia, vomiting, constipation, diarrhea, gastroesophageal reflux, hypoglycemia, and renal stones. These results are in line with what has been presented above. Thus, with adequate monitoring of side effects, KDs are safe and efficacious in infants.

**Table 1 children-09-01372-t001:** Side effects observed in ketogenic diets in different trials (table adapted from Acharya et al. 2021 with permission from author [59]).

Author	Different Types of Ketogenic Diet *	Side Effects/ Complication of Ketogenic Diet besides Renal Stones
Holler A et al. [118]	Classical KD and MAD	Constipation, increased bromine level (3.2%)
Attar H et al. [119]	MAD	NR
Felix et al. [120]	MAD	Weight loss, hyperlipidemia
Mak et al. [121]	KD – Protein + carb (<19%) of caloric requirementsMCT 60–70% of caloric requirements	Weight loss (46%), diarrhea (38%), bad temper (7.6%), abdominal cramps (15%), nausea (15%), bad body smell (7.6%)
Rios et al. [122]	KD 4:1 (1.5:1 to 4.5:1)	Nausea and vomiting (26.3%), hypercholesterolemia (64.7%), anorexia (31.8%), constipation (40.9%), symptomatic acidosis (9.09%), carnitine deficiency (9.09%)
Sharma et al. [123]	Classical KD 3:1 or 4:1	Vomiting (75%), asymptomatic hypocalcemia, constipation (75%), weight loss, hypoalbuminemia
Kang et al. [30]	Classical KD 4:1	Dehydration, GI discomfort, hyperlipidemia, hyperuricemia, symptomatic hypoglycemia, lipoid aspiration pneumonia, hypoproteinemia, hypomagnesemia, repeated hyponatremia
Wibisono et al. [124]	Classical KD, MCT, MAD	Constipation, hypertriglyceridemia, hypercholesterolemia, diarrhea, lethargy, iron deficiency, GERD, vomiting, hypoglycemia
Simm et al. [77]	KD 2:1 to 4:1	Osteopenia, fracture
Guzel et al. [125]	KD 2.5:1 and 4:1	Hyperlipidemia (50.8%), selenium deficiency (26.9%), constipation (26.2%), sleep disturbances (20%), hyperuricemia (3%), hepatic effects (2.6%), hypoproteinemia (2.6%), hypoglycemia(1.5%)
Hassan et al. [40]	Classic 4:1 KD or MCT diet	Constipation (85%), gall bladder stone (1.9%), hyponatremia (1.9%)
Takeoka et al. [126]	KD not specified	Nausea/vomiting (7%), irritability (7%), lethargy (21%), sedation (14%)
Kossoff et al. [56]	KD 3:1 to 4:1	NR
Kossoff et al. [127]	KD 3:1 to 4:1	Sedation (27%), rash, irritability
Kossoff et al. [128]	KD 3:1 to 4:1	Severe GERD (13%), hip dislocation (0.4%)
Kang et al. [129]		Dehydration, GI discomfort, hyperlipidemia, hyperuricemia, hypoglycemia
Mackay et al. [130]	Classical KGD 3:1 to 4.2:1	Asymptomatic hypoglycemia (24%), poor linear growth (20%), hyperlipidemia (16%), vomiting (12%), hypocarnitinemia (8%), hypercalciuria (8%), constipation (8%), osteopenia (4%), pancreatitis (4%), diarrhea (4%)
Groesbeck et al. [131]	60.7% on classical KD7% MAD, 32% other KD	Fractures (21.4%), hyperlipidemia (7%), constipation (53%)
Sampath et al. [57]	KGD 3:1 (56%) or 4:1	NR
Caraballo et al. [132]	KD	GI side effects (30.5%), hyperlipidemia (9.7%), weight gain (2.3%), hypocarnitinemia (3.7%), hypercalciuria (6.9%), hypoglycemia (5.5%), dehydration (6.4%)
Dressler et al. [61]	KD 3:1,4:1 or 2.5:1	Carnitine deficiency (13%), growth deficit (5.2%), weight gain (1.7%), hypertriglyceridemia (29.5%), hypercholesterolemia (10.4%)
Hallbook et al. [133]	KD 3:1 or 4:1 ratio	Hyperlipidemia (6%), bone fractures (0.9%)
Lim et al. [134]	NR	GI side effects (nausea, vomiting, and constipation), inadequate weight gain or significant weight loss, ketoacidosis, hepatotoxicity, renal dysfunction, sinus tachycardia, osteoporosis
McNally et al. [60]	KGD unspecified	NR
Park et al. [135]	KD 4:1 (87.5%), KD 3:1 (12.5%)	Regurgitation, constipation, aspiration, hypertriglyceridemia, hypoproteinemia, nausea, vomiting
Draaisma et al. [136]	Classic KD (67.6%), MCT diet (2.9%)MAD (19.1%), LGID (7.4%), others (1.5%)	Decrease in BMD 0.22 Z-score/year
Roehl et al. [137]	Modified ketogenic diet−15 gm carb vs 50 gm carb diet	Constipation 9%
Lambrechts et al. [138]	KD	GI side effects (30%)

* KD: Ketogenic Diet, MAD: Modified Atkins Diet, MCT: Medium Chain Triglyceride Diet, LGID: Low Glycemic Index Diet.

## 3. Conclusions

Overall, KDs are a well-tolerated and efficacious therapy modality for refractory idiopathic epilepsy and genetic epileptic syndromes. However, there are certain potential side effects with the use of KDs that one must be aware of. The primary risk factor for developing many side effects appears to be a higher fat to carbohydrate ratio. Therefore, this ratio should be adjusted under the supervision of a pediatric neurologist to attain a tolerable and therapeutic KD. Dieticians are invaluable members of the team when using a KD as they can help design a diet that will meet therapeutic parameters while maximizing patient satisfaction through rotation of food size, color and aroma [139].

Finally, a number of potential side effects tend to be transient, making the KD an efficacious and safe treatment option for epilepsy when patients are cared for by an experienced medical team. Table 2 summarizes common potential complications according to risk factors such as family history and preexisting conditions. Furthermore, the appropriate clinical monitoring and interventions to reduce these complications are also listed.

## Figures and Tables

**Table 2 children-09-01372-t002:** Potential complications according to high-risk patient populations, with the corresponding appropriate clinical monitoring and interventions.

Potential Complication	High Risk Patient Population	Clinical and Investigational Monitoring	Possible Intervention to Reduce Complication
Nausea, vomiting, worsening of reflux disease	Young infants, Multiple AED, GERD, Hypotonia	- Serial urine ketones, blood glucose - BHB - Weight - GI consultation	- Anti-reflux medication (e.g., PPI) - Anti-nausea medication (e.g., Ondansetron) - Treat constipation if necessary - Positioning while feeding - Changing formula type/consisting in G-tube and changing time of various medications in relation to diet administration - Small bolus or continuous feeding
Constipation	Prior h/o constipation, Autism, Hypotonia and GDDYoung infants	- Formal GI consultation and serial monitoring	- Senna - Miralax - Feeding schedule - Biofeedback therapy
Weight loss	Low BMI before KD initiation, Genetic epilepsy, GDD, Use of ASM such as Topiramate	- Serial weight monitoring	- Use of higher calories and MCT-based diet to maintain ketosis and improve weight - Avoid Topiramate if possible
Osteopenia	Poor bone density, Low Vit D, Prior h/o fractures, h/o genetic condition such as Osteogenic imperfecta	- Serial Vit D level - Serial Dexascan - Endocrine evaluation	- Vit D replacement - Adequate sunlight exposure - Biphosphonates under endocrinology supervision
Persistently Low ASM levels	Use of Multiple ASM, including enzyme-inducing ASM	- Serial ASM monitoring	- Optimize dose as tolerated for low ASM levels - Avoid multiple ASMs that interact with each other’s metabolism, and avoid enzyme inducing ASMs
Hypoglycemia	Infants, Low BMI, High KD ratio H/o vomiting and poor feeding tolerance, Surgical procedures	- Serial blood glucose monitoring q6H and PRN - Small and continuous feeding	- Small quantity of orange juice PRN - Lower KD ratio if necessary
Carnitine deficiency	Prior low carnitine levels, Use of High KD ratio, Use of ASM such as Valproic acid	- Serial monitoring levels of Carnitine profile - Monitor clinical symptoms such as fatigue, tiredness, abnormal LFTs	- Supplement Levocarnitine 50–100 mg/kg/day in tablet form, if symptomatic
Hyperlipidemia	Family h/o hyperlipidemias, High BMI, Type II DM, High KD ratio	- Serial fasting lipid profile	- Low use of trans and saturated fat KD - Increase polyunsaturated and monounsaturated fat - Carnitine replacement - Physical exercise
Renal stones	Infants, H/o prematurity, H/o renal malformation, High KD ratio, Concurrent use of ASM such as Topiramate, Family h/o renal stones	- Serial urine calcium/creatinine - Serial renal USG - Serial BHB level - Nephrology and urology consultation before and after initiation of KD	- Avoid ASMs such as Topiramate - Lower KD ratio - Use urine alkalization medications such as Polycitra-K^®^
Acidosis	Infants, High KD ratio, Concurrent use of ASM such as Topiramate	- Serial BMP	- Use of baking soda or sodium bicarbonate tablets or - Polycitra-K^®^ (2 meq/kg/day) - Lower KD ratio if necessary

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
