# Peer review of "A Review of the Multi-Systemic Complications of a Ketogenic Diet in Children and Infants with Epilepsy"

_children, 2022, doi:10.3390/children9091372_

Round 1

Reviewer 1 Report

The authors present an interesting evaluation of the ketogenic diet, but:

Line 92: it should be "rely on ketone bodies and fat (SNC uses only ketones other cells can use fatty acid.

Line 110: LGIT is not properly a KD, so it could be cited, but it should be pointed out that it is not a KD

Line 155-158 to say that at least 40% of subjects have intestinal problems is exaggerated, and above all, it should be related to how the diet is organized.

Line 165: enteropathy is a generic condition; it should be better seen what was the exact cause, as reported in line 189

- Cardiac and renal, it should be pointed out that both conditions could be managed with the support of basic salts (did they use?)

In the conclusion, it should be considered that many reported side effects depend on how the diet is organized. This does not mean neglecting them; indeed, they should be considered to improve the diet proposal.

Finally, the anti-inflammatory effect of KD should also be emphasized as a possible adjuvant action, particularly in chronic diseases.

Reviewer 2 Report

The authors say that this is a review manuscript using PUBMED and MEDLINE. However, it seems that the authors misunderstand a systematic review and a narrative review and mixed up them.

Major issues

#1. Please remove redundant explanation for epilepsy.

#2. Please use a professional proofreading service. There are duplicated abbreviations.  

#3. Please simplify the introduction part. Since readers who are interested in this study would be specialists who know their mechanisms.

#4. Method part is too rough. How did you search for the literatures? Key words? AND and OR should be explained.

#5. How many literatures did you find? Among these, which inclusion criteria, and exclusion criteria reach to the results?

Minor issues

#1. Even in abstract, first use of abbreviation need full words and its abbreviation.

#2. The definition of intractable epilepsy that is adequate 2 or more medications parts needs citation.

Round 2

Reviewer 1 Report

Line 134, you left five types of KD, including LGIT; you should write four and consider LGIT such possible effective but off-topic in the manuscript.

In the conclusion, you should resume better the possible side effects, pointing out those that may depend on the specific diet (constipation, nausea, and the suchlike) or those from conditions, sometimes pre-existing, to be better monitored; as the former can be highly reduced if the person prescribing the diet has the necessary experience in the field. A table could be the best option

Reviewer 2 Report

The authors majorly revised the manuscript as a form of a narrative article. I just have a minor recommendation.

Minor issues

#1.  I found several mistakes, typos, etc. Please use a professional proofreading service.
